# Identification of a pathway for electron uptake in *Shewanella oneidensis*

Annette R. Rowe [1,5✉], Farshid Salimijazi [2,5], Leah Trutschel[1,5], Joshua Sackett[1], Oluwakemi Adesina[3], Isao Anzai[3], Liat H. Kugelmass[3], Michael H. Baym [4] & Buz Barstow [2✉]

Extracellular electron transfer (EET) could enable electron uptake into microbial metabolism for the synthesis of complex, energy dense organic molecules from $CO_2$ and renewable electricity[1–6]. Theoretically EET could do this with an efficiency comparable to $H_2$-oxidation[7,8] but without the need for a volatile intermediate and the problems it causes for scale up[9]. However, significant gaps remain in understanding the mechanism and genetics of electron uptake. For example, studies of electron uptake in electroactive microbes have shown a role for the Mtr EET complex in the electroactive microbe *Shewanella oneidensis* MR-1[10–14], though there is substantial variation in the magnitude of effect deletion of these genes has depending on the terminal electron acceptor used. This speaks to the potential for previously uncharacterized and/or differentially utilized genes involved in electron uptake. To address this, we screened gene disruption mutants for 3667 genes, representing ≈99% of all nonessential genes, from the *S. oneidensis* whole genome knockout collection using a redox dye oxidation assay. Confirmation of electron uptake using electrochemical testing allowed us to identify five genes from *S. oneidensis* that are indispensable for electron uptake from a cathode. Knockout of each gene eliminates extracellular electron uptake, yet in four of the five cases produces no significant defect in electron donation to an anode. This result highlights both distinct electron uptake components and an electronic connection between aerobic and anaerobic electron transport chains that allow electrons from the reversible EET machinery to be coupled to different respiratory processes in *S. oneidensis*. Homologs to these genes across many different genera suggesting that electron uptake by EET coupled to respiration could be widespread. These gene discoveries provide a foundation for: studying this phenotype in exotic metal-oxidizing microbes, genetic optimization of electron uptake in *S. oneidensis*; and genetically engineering electron uptake into a highly tractable host like *E. coli* to complement recent advances in synthetic $CO_2$ fixation[15].

[1] Department of Biological Sciences, University of Cincinnati, Cincinnati, OH, USA. [2] Department of Biological and Environmental Engineering, Cornell University, Ithaca, NY, USA. [3] Department of Chemistry, Princeton University, Princeton, NJ, USA. [4] Department of Biomedical Informatics, Harvard Medical School, Boston, MA, USA. [5] These authors contributed equally: Annette R. Rowe, Farshid Salimijazi, Leah Trutschel. ✉email: annette.rowe@uc.edu; bmb35@cornell.edu

Electromicrobial production technologies aim to combine the flexibility of $CO_2$-fixing and $C_1$-assimilating microbial metabolism for the synthesis of complex, energy-dense organic molecules from $CO_2$ and renewable electricity[1–6]. Already, the Bionic Leaf device has demonstrated that technologies of this class could dramatically exceed the efficiency of photosynthesis[7,8]. However, while highly efficient at lab scale, the Bionic Leaf relies on $H_2$ oxidation to transfer electrons from the electrode to microbes, and the low solubility of $H_2$ in water would pose a significant challenge for scale-up of this and related technologies[9].

Extracellular electron uptake (EEU) as an electron source for metabolism could allow engineers to circumvent the scale-up limitations of $H_2$ oxidation. Naturally occurring electro-autotrophic microbes can produce acetate and butyrate from $CO_2$ and electricity with Faradaic efficiencies exceeding 90%[16]. Furthermore, theoretical analysis suggests that the upper-limit efficiency of electromicrobial production of biofuels by EEU could rival that of $H_2$-mediated systems[9]. However, naturally occurring electroactive organisms capable of EEU suffer from multiple technical drawbacks. Most notably, they have a low-tolerance to high-osmotic-strength electrolytes, requiring the use of electrolytes that confer low electrochemical cell conductivity and thus a low overall energy efficiency. Additionally, they have a poor ability to direct metabolic flux to a single product more complex than acetate or butyrate[16]. Correcting these problems to take full advantage of EEU's potential by genetic engineering[17] will require extensive knowledge of the genetics of EEU.

Growing evidence suggests that the model electroactive microbe S. oneidensis can couple EEU to the regeneration of ATP and NADH, both essential precursors for biosynthesis[13], by reversal of its extracellular electron transfer (EET) pathway (Fig. 1), making it an attractive chassis organism for electro-microbial production. However, EEU machinery in S. oneidensis appears to involve more than just operating the well-characterized EET machinery in reverse[13,18]. EEU in S. oneidensis can link cathodic current with multiple terminal electron acceptors, including oxygen, which draws into question how electrons transfer between canonically discrete electron-transport chains. Finding this machinery has been hindered by the lack of high-throughput assays for electron uptake and the challenge of developing screens for non-growth-related phenotypes. Even with recent advances in high-throughput electrode arrays[19], searching through the thousands of genes in even a single microbial genome by direct electrochemical measurements remains impractical.

To address this, we developed a rapid colorimetric assay to screen all 3667 members of the S. oneidensis whole-genome-knockout collection[20,21] (covering ≈99% of all nonessential genes) and characterize the genetics of EEU. The assay relies upon oxidation of the reduced form of the redox dye anthra(hydro)quinone-2,6-disulfonate ($AHDS_{red}$ for the reduced form and $AQDS_{ox}$ for the oxidized form) and is coupled to reduction of the anaerobic terminal electron acceptors fumarate and nitrate[22–24] (Figs. 2 and S1). While $AHDS_{red}/AQDS_{ox}$ redox dye assays are not a perfect proxy for EEU and EET, they are capable of identifying many components of the S. oneidensis EET machinery[20]. While $AHDS_{red}/AQDS_{ox}$ does appear to be able to enter the cell, it also appears to be rapidly pumped out by a TolC efflux pump[25]. We suspect these results in a lower concentration of $AHDS_{red}/AQDS_{ox}$ in the interior of the cell than in the exterior solution. As a result, cell-surface proteins, like the well-known Mtr EET complex, are responsible for transferring a detectable fraction of electrons to $AQDS_{ox}$ (ref. [20]). Thus, we believed it was reasonable to assume that the $AHDS_{red}$ oxidation assay could detect genes involved in EEU. To ensure that genes are involved in EEU with solid surfaces, a subset was tested in electrochemical systems, the gold standard for measuring EEU[26,27].

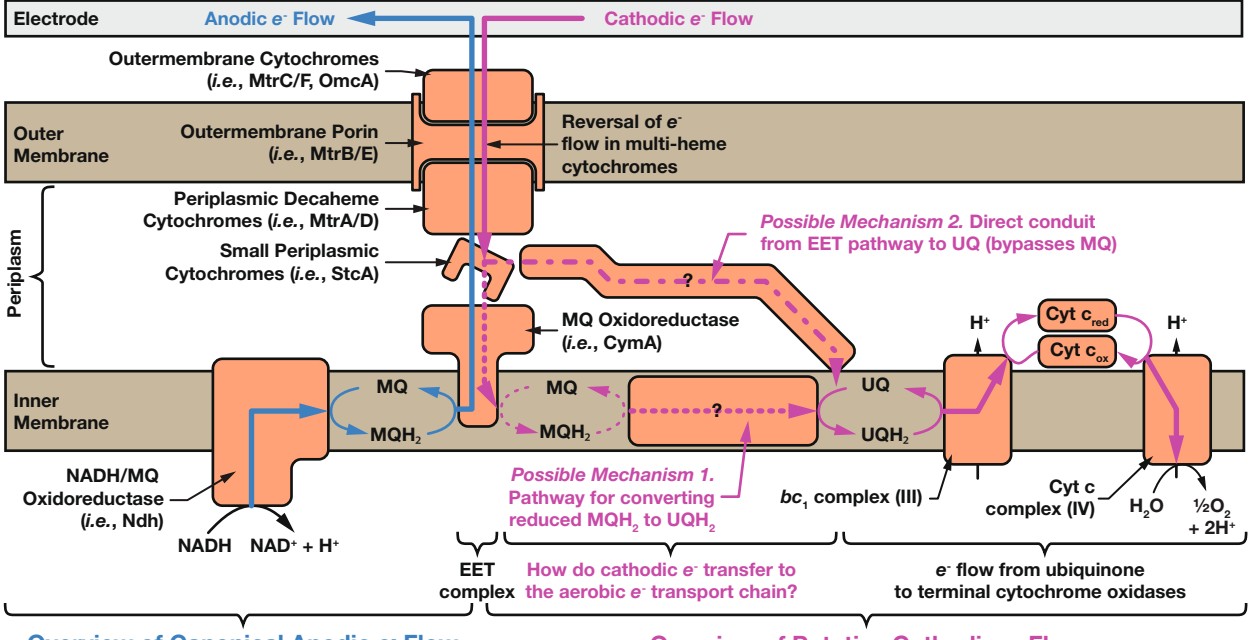

**Fig. 1 Electron uptake in the model electroactive microbe *Shewanella oneidensis* MR-1 cannot be fully explained by reversal of its extracellular electron-transfer pathway.** The canonical anodic extracellular electron-transport (EET) pathway for electron deposition is shown in light blue and the putative cathodic extracellular electron-uptake (EEU) pathway is shown in pink. Known electron-transfer pathways are denoted with solid lines, while speculated transfer pathways are shown as dashed lines. Two possible mechanisms for transfer of cathodic electrons from the Mtr EET complex to the ubiquinone pool and onto terminal cytochrome oxidases are highlighted. We speculate that two of the proteins identified in this work (SO_0400 and SO_3662) could form part of possible mechanism 2.

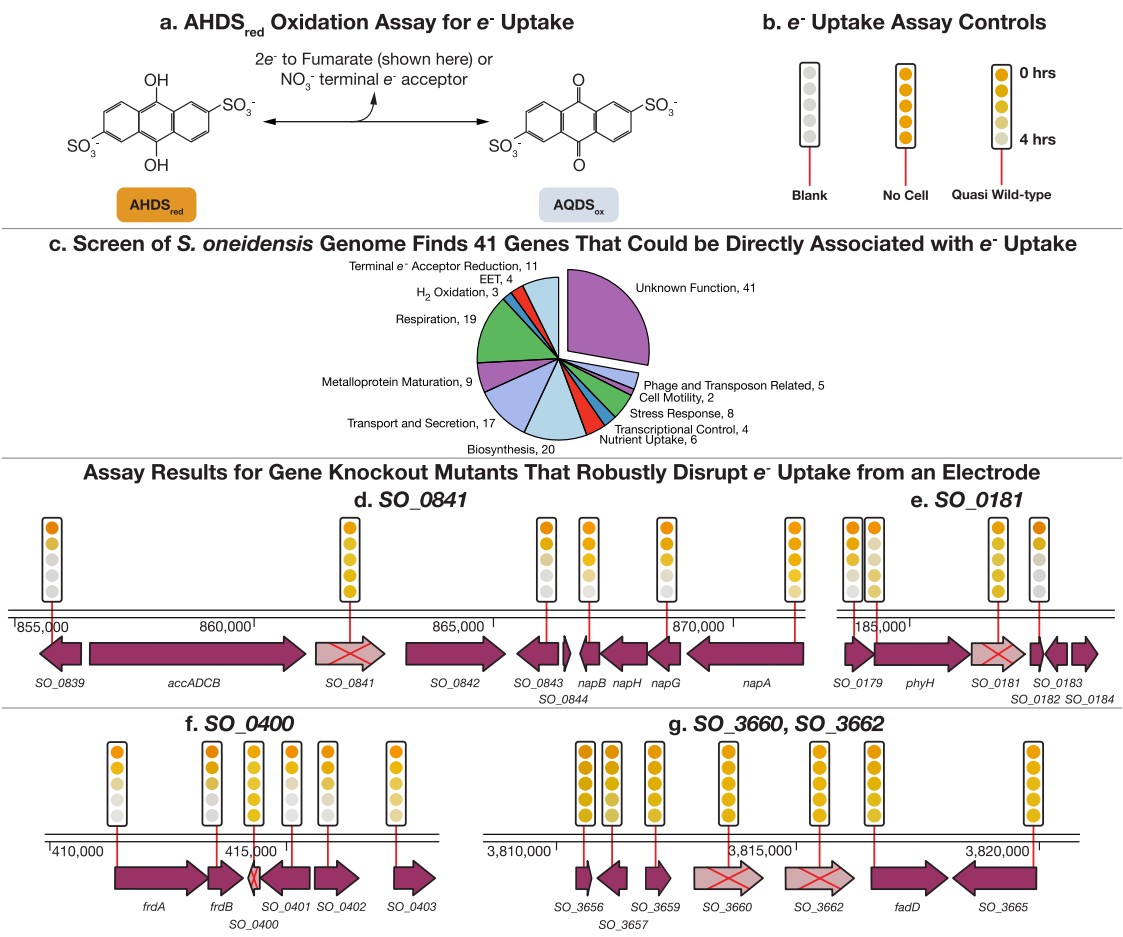

**Fig. 2 A genome-wide screen of *S. oneidensis* finds 149 genes that disrupt electron uptake.** All 3712 members of the *S. oneidensis* whole-genome-knockout collection were screened for electron uptake capability with AHDS$_{red}$ oxidation assays, either fumarate or nitrate as a terminal electron acceptor. In total, 149 genes disrupt AHDS$_{red}$ oxidation with fumarate, nitrate and in many cases both as a terminal electron acceptor (note, only one terminal electron acceptor is used at once). **a** AHDS$_{red}$/AQDS$_{ox}$ redox reaction is used as a proxy for extracellular electron uptake. AHDS$_{red}$ changes color from orange to clear when oxidized. Electrons are transferred to either a fumarate or nitrate terminal electron acceptor by *S. oneidensis*. **b** Blank, no-cell and quasi-wild-type (transposon mutants that contain a kanamycin cassette but have no effect on AHDS$_{red}$ oxidation) controls. The color of the AHDS$_{red}$ dye is recorded photographically and displayed at 1-hour intervals after the start of the experiment by a series of colored circles above each gene. Further information on this assay can be seen in Fig. S1 and "Materials and Methods". Data shown use fumarate. **c** The electron-uptake assay associates 149 genes with electron uptake. Electron uptake failure can be explained in 108 cases, but in 41 cases, it fails for unknown reasons, implicating these genes in an uncharacterized electron-uptake process. Full screening results and functional categorizations are shown in Supplementary Data 1. **d–g** AHDS$_{red}$ oxidation assay results are shown for selected mutants containing deletions of genes highlighted in this article that robustly disrupt electron uptake from a cathode (the selection process for these five mutants is shown in Fig. 3) (pink arrow with a red cross through the center) along with gene-disruption mutants for the surrounding genes (purple arrow, with a red line indicating the location of the transposon insertion). All time courses are from experiments using fumarate as a terminal electron acceptor.

## Results and discussion

**High-throughput electron uptake screen finds 41 genes with unknown function.** We identified mutants in 149 coding and intergenic regions in the *S. oneidensis* genome that slowed or eliminated AHDS$_{red}$ oxidation with fumarate, nitrate, or both terminal electron acceptors (Supplementary Data 1). While O$_2$ is the most useful terminal electron acceptor for electromicrobial production due to its high redox potential and enormous availability, we were unable to design a reliable high-throughput assay that used it. AHDS$_{red}$ is exquisitely sensitive to oxidation by O$_2$, forcing us to use fumarate and nitrate, both of which cannot directly oxidize AHDS$_{red}$, but instead require *S. oneidensis* as an intermediate.

Among 149 hits, 18 mutants were slow or failed at oxidation of AHDS$_{red}$ in only the fumarate assay, 50 mutants, in only the nitrate assay, and 81 mutants in both assays. In total, 109 of these

mutants were grouped by gene annotation into functional categories that satisfactorily explain the slowing or failure of AHDS$_{red}$ oxidation (Fig. 2c). For example, disruption of the periplasmic fumarate reductase (δ*fccA*; we refer to transposon-disruption mutants with δ, and gene-deletion mutants with Δ) eliminates AHDS$_{red}$ oxidation when using fumarate as a terminal electron acceptor. Detailed time courses of AHDS$_{red}$ oxidation for selected anticipated hits from the genome-wide screen are shown in Fig. S2. Of note, 41 of the AHDS$_{red}$ oxidation-deficient mutants could not be assigned to an established functional category, suggesting that their function might be more directly involved in electron uptake (Fig. 2c). AHDS$_{red}$ oxidation time courses for knockout mutants, where we later observed a cathode phenotype, are shown in Fig. 2d–g, along with those for mutants with disruptions in adjacent genes. Detailed time courses for these mutants are shown in Fig. S3.

**Electrochemical measurements confirm robust EEU phenotype of five unknown function mutants**. We selected 17 of the 41 "unknown function" *S. oneidensis* $AHDS_{red}$ oxidation-deficient mutants for further on-electrode testing. These mutants were chosen for annotations that indicated possible redox activity (e.g., *δSO_3662*), interaction with the quinone pool (e.g., *δSO_0362*, *δSO_0400*), along with mutants with no functional annotation. To exclude genes involved in solely in terminal electron-acceptor utilization (the very end of the electron-transport chain), we used a different terminal electron acceptor ($O_2$) than in the $AHDS_{red}$ oxidation screen. The use of $O_2$ also ensures that the genes identified are part of the overlapping electron-uptake pathway, rather than previously unidentified components of fumarate/ nitrate reduction. We confirmed this using *δfccA*, *δnapA* and *δnapG* as negative controls as these genes encode anaerobic terminal reductases that we did not expect to disrupt electron uptake using $O_2$ as a terminal electron acceptor. We also selected three positive control mutants of genes known to be involved in EET (*δcymA*, *δmtrA* and *δmtrC*) and one expected negative control based on $AHDS_{red}$ oxidation screen results (*δSO_0401*). *δSO_0401* was chosen as it is adjacent to a hit (*δSO_0400*) in the $AHDS_{red}$ oxidation assay, but does not itself produce an oxidation phenotype.

Biofilms of each of the mutants were grown on ITO working electrodes in a three-electrode bio-electrochemical system[13]. For analysis of electron uptake, the working electrodes were poised at −378 mV vs. the standard hydrogen electrode (SHE). Significant negative currents (i.e., electrons flowing from the working electrode to the biofilm/solution) were only observed in the presence of $O_2$ as a terminal electron acceptor. To quantify the amount of negative current due to biological vs. nonbiological processes, the electron-transport chain was inhibited at the end of each experiment with the ubiquinone mimic, Antimycin A and the remaining abiotic current was measured (Fig. S4). Each mutant was tested in at least three replicate experiments.

Most of the 17 mutants of unknown function demonstrate a limited-to-modest change in average electron uptake from the working electrode (Figs. 3a, S5a, S5c, S5d, and Table S1). As expected, mutants that disrupt components of the well-known Mtr EET complex produce significant reductions (*p* value < 0.05) in electron uptake, except for *cymA*[10,13]. Though *cymA* was previously shown to be important under anaerobic cathodic conditions[10], only a small reduction in electron uptake was noted under aerobic conditions, consistent with previous results[13]. It is plausible that the other unknown genes tested that did not generate a cathodic phenotype play a previously uncharacterized role in one of the other subcategories highlighted in the $AHDS_{red}$ assay rather than electron uptake, such as the reduction of fumarate or nitrate, as opposed to $O_2$.

Disruption mutants in four genes that code for proteins that are of unknown function (*δSO_0181*, *δSO_0400*, *δSO_0841* and *δSO_3660*) demonstrated a highly significant reduction in electron uptake (*p* value < 0.05) (Fig. 3a). At the time of collecting data for Fig. 3a, b, we were unable to retrieve *δSO_03662* from the *S. oneidensis* whole-genome-knockout collection. To further verify that the reductions in current seen in *δSO_0181*, *δSO_0400*, *δSO_0841* and *δSO_3660* were due to the loss of the disrupted gene and not due to any polar effects, we made a clean deletion for each gene, which all demonstrated a decreased electron-uptake phenotype compared with wild-type (Fig. 3c). We also constructed a clean deletion mutant for *SO_3662* (*ΔSO_3662*) to collect data for Fig. 3c. *ΔSO_3662* (annotated as an inner membrane ferredoxin), demonstrated both an $AHDS_{red}$ oxidation phenotype (Fig. 2g and Fig. S3) and a cathodic phenotype (Fig. 3c). Furthermore, complementation of the

knockout mutants with a plasmid encoding the deleted gene restored electron uptake function in all mutants (Fig. 3c).

We were later able to retrieve *δSO_3662* mutant and retested it in the $AHDS_{red}$ oxidation assay prior to submitting this article. In addition, while retesting *δSO_3660* and *δSO_3662*, we tested gene-disruption mutants for the surrounding genes and found that these also disrupted $AHDS_{red}$ oxidation (Fig. 2g). However, as of the time of writing, we have not yet validated these mutants by electrochemical measurements.

**Electron uptake disruption is not due to biofilm formation failure**. Electron-uptake disruption in these five gene-deletion mutants cannot be explained by changes to biofilm production or cell growth rate (Tables S2 and S3). We used protein quantification to ensure that the cathode phenotypes we observed were not caused by deficiencies in biofilm formation due to direct interference with biofilm formation or failure of anode conditioning prior to cathodic current measurement. In an earlier work[13], we screened mutants of the known EET pathway (known anodic phenotypes) for cathodic phenotypes. We imaged all the biofilms, performed cell counts, and quantified the total protein from the electrode biofilms[13]. We observed no evidence of differences between the biofilm-formation capability of the mutants we screened compared with wild-type. While protein measurements were variable, they were no less so than microscopy-count data. Thus, we adopted protein quantification to measure biofilm-cell abundance.

No statistically significant difference in on-electrode protein abundance was seen for any of the mutants tested (Table S1). Furthermore, no statistically significant difference in growth rate between wild-type *S. oneidensis* and any mutant was observed in minimal media under aerobic or anaerobic conditions, with the exception of a longer lag time observed for complementation mutants containing an extra plasmid and grown in the presence of chloramphenicol and/or kanamycin (Table S2 and Fig. S6).

**Four of the five genes are only involved in electron uptake**. In addition to electron uptake, we also analyzed the extracellular electron deposition of each of the 24 mutants selected for on-electrode testing (Fig. 3b). Of the five EEU genes identified here (*SO_0181*, *SO_0400*, *SO_0841*, *SO_3660* and *SO_3662*), only disruption of *SO_0841* significantly reduces both electron uptake and deposition. The *δSO_0841* disruption mutant produces half the positive current of wild-type *S. oneidensis*, similar to the effect of disruption of *mtrA* and *mtrC* (Table S1 and Fig. 3b). Interestingly, no growth phenotype was observed for cells grown on soluble (iron citrate) or insoluble iron (iron oxyhydroxide), suggesting that *SO_0841* does not play a significant role in EET to minerals.

**Sequence analysis suggests gene functions and shows electron uptake genes are widely distributed across species**. Phylogenetic analysis of the five EEU genes identified here suggests that they are broadly conserved across *Shewanella* species and across numerous clades of the *Gammaproteobacteria*. Phylogenetic trees are shown in Figs. S7–S12. Metadata for trees are attached in Supplementary Data 2.

*SO_0841*. Sequence analysis suggests that *SO_0841* is involved in cell signaling. *SO_0841* encodes a transmembrane protein with a 250-amino-acid long periplasmic region and a 250-amino-acid long cytoplasmic domain containing a GGDEF c-di-GMP signaling domain. As a phenotype was only observed on electrodes, and not iron minerals, the role of this protein in traditional EET remains unclear. The GGDEF domain is often used for regulation

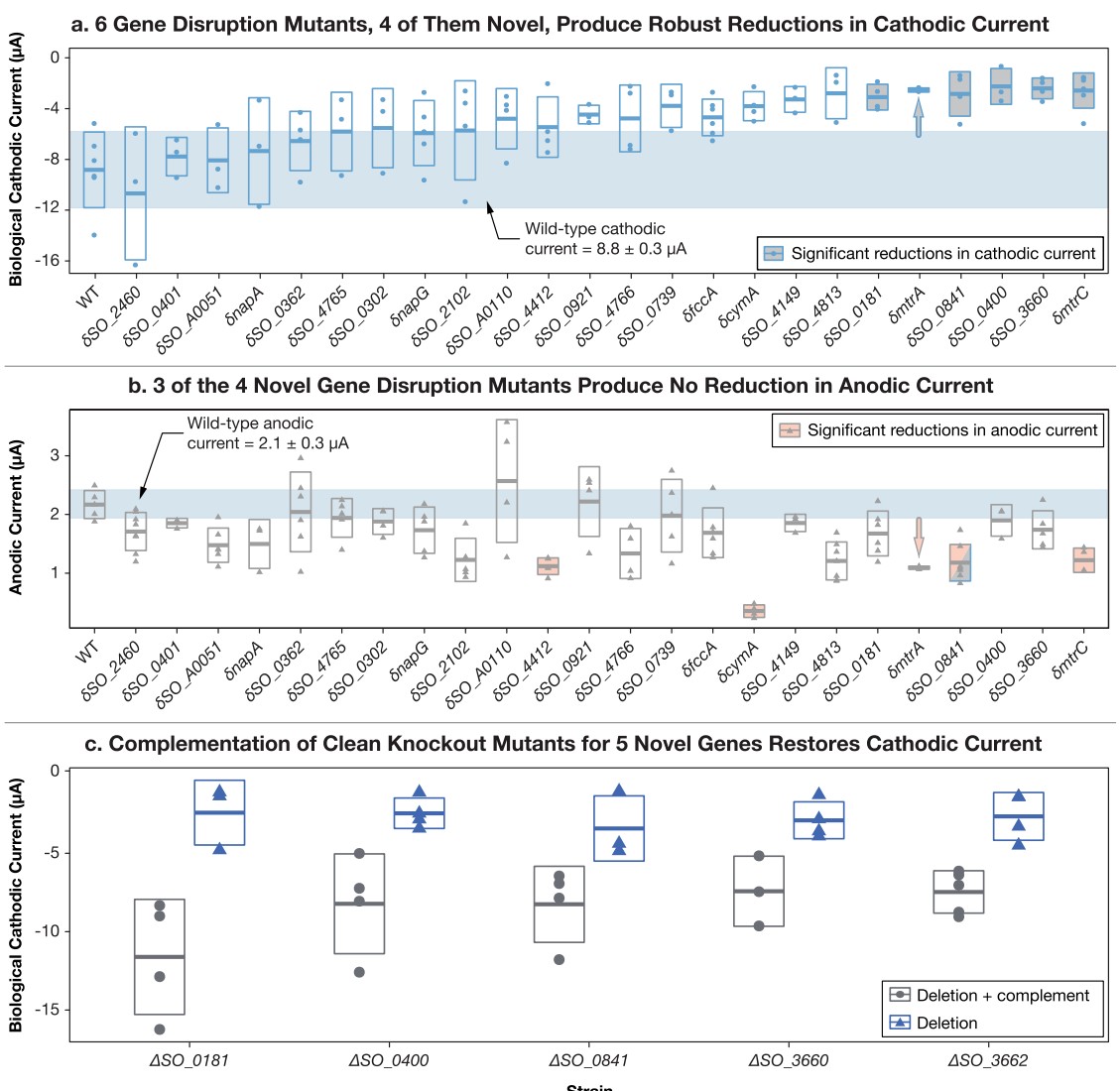

**Fig. 3 Electrochemical measurements confirm robust extracellular electron-uptake phenotype for 5 *S. oneidensis* mutants identified by our high-throughput screen ($\delta$SO_0181, $\delta$SO_0400, $\delta$SO_0841, $\delta$SO_03660 and $\delta$SO_03662; ($\delta$ indicates a transposon-insertion mutant)).** $\delta$SO_03662 was implicated in electron uptake by the $AHDS_{red}$ oxidation assay. However, at the time of collecting data for panels **a**, **b**, we were unable to retrieve $\delta$SO_03662 from the *S. oneidensis* whole-genome-knockout collection, so later constructed a clean deletion mutant to collect data for panel **c**. Hence, we talk about four mutants in panels **a**, **b**, but 5 in panel **c**. We were later able to retrieve $\delta$SO_3662 mutant and retested it in the $AHDS_{red}$ oxidation assay prior to submitting this article. The $\delta mtrA$ control mutant in panels **a**, **b** shows significant derivation from wild type, but the current range is too small to be drawn clearly, so is marked with arrows. **a** We measured the cathodic biological current for 18 *S. oneidensis* mutants that produced $AHDS_{red}$ oxidation failure for unknown reasons, control mutants ($\delta fccA$, $\delta mtrA$, $\delta mtrC$, and $\delta napG$), and wild type (WT). In total, 6 gene-disruption mutants, four of them previously uncharacterized, produced robust reductions in cathodic current. The region corresponding to wild-type currents and one standard deviation is shaded in blue. ANOVA results indicated that there was a significant variation in means of the strains from one another ($F$-value = 5.94; Pr($>F$) = $2.34 \times 10^{-9}$), while individual comparisons specifically revealed $\delta$SO_0181, $\delta mtrA$, $\delta$SO_0841, $\delta$SO_0400, $\delta$SO_3660, and $\delta mtrC$ to all display significantly lower cathodic currents ($p < 0.05$) than WT. **b** However, three of the six robust electron-uptake mutants (three of the four previously uncharacterized disruption mutants) do not have a significant effect on anodic current production (electron deposition), the well-characterized phenotype of *S. oneidensis*. The range of wild-type current values and a standard deviation from the mean is shaded in blue. Analysis of anodic current values ($F$-value = 5.801, Pr($>F$) = $2.10 \times 10^{-10}$) revealed that of the four previously uncharacterized disruption mutants, only $\delta$SO_0841 shows anodic current production different from WT ($p < 0.05$). **c** Gene deletion (indicated by $\Delta$) confirms the electron uptake phenotype of five mutants. Complementation of the deleted gene restores electron uptake phenotype. Further information on electrochemical methods can be seen in Materials and methods. Data are available in ref. [53].

of biofilm formation and cell motility[28]; however, disruption of *SO_0841* had no on impact on biofilm or cell morphology (Table S1 and Fig. S7).

This suggests a more specific role for SO_0841 in on-electrode EET. SO_0841 also has a broad distribution of homologs ranging across the *Proteobacteria* (Fig. S8). Though electron-uptake phenotypes have not been tested in a wide range of organisms, homologs of SO_0841 are found in electrochemically active microbes with the capacity for EEU, including *Mariprofundus ferrooxydans*[29], *Idiomarina loihiensis*[30] and *Marinobacter* species[30]. Thus, we speculate that this gene may have a conserved role in electron uptake.

The remaining four genes identified all play significant roles in electron uptake, but have no detectable role in electron deposition (Fig. 3a, b).

*SO_0181.* SO_0181 is predicted to be membrane-associated and contains a putative nucleoside triphosphate-binding and/or hydrolase domain (suggesting that it interacts with ATP or GTP). Furthermore, SO_0181 is located immediately upstream of *phyH* which encodes an uncharacterized putative oxidoreductase (part of a family of bacterial dioxygenases with unconfirmed activity in *S. oneidensis*), which also demonstrated an electron-uptake phenotype in the AHDS$_{red}$ oxidation screen (Supplementary Data 1 and Fig. 2e).

There is little bioinformatic support for a direct role of SO_0181 in redox chemistry. However, a role associated with activating or modifying the other redox-active proteins involved in EEU seems feasible but needs to be further investigated.

A phylogenetic tree constructed from the 200 closest relatives of SO_0181 shows a distinct clade specific to the *Shewanellae* (90–100% sequence identity) (Fig. S9). Closely related clades of SO_0181 are found in the *Pseudomonas*, *Cellvibrio* and *Hahella* genera of the *Gammaproteobacteria*, though homologous gene clusters are also seen in *Beta-* (*Delftia* and *Acidovorax*) and *Delta-proteobacterial* (*Archangium* and *Cystobacter*) lineages. Notably, several of the genera with close homologs of SO_0181, including *Delftia* and *Pseudomonas*, have EET-capable representatives within them.

*SO_0400.* SO_0400 belongs to a superfamily of quinol-interacting dimeric monooxygenses (dimeric α–β-barrel superfamily SSF54909). Of the homologs of known function, SO_0400 is most closely related to the YgiN quinol monooxygenase in *E. coli*[31]. Structural analysis of YgiN suggests that it interacts with the semiquinone state of quinols and suggests the existence of an unusual quinone redox cycle in *E. coli*[31]. Additionally, deletion of *ygiN* in *E. coli* inhibits the transition between aerobic and anaerobic growth[32]. Interestingly, deletion of SO_0400 does not inhibit the transition between aerobic and anaerobic growth conditions (and vice versa) in *S. oneidensis* (Fig. S6). Furthermore, deletion of SO_0400 did not affect the sensitivity of *S. oneidensis* to oxygen-free radicals using a disc-diffusion assay with hydrogen peroxide (data not shown). These data suggest a possible previously uncharacterized function in this quinol monooxygenase.

SO_0400 has a very broad distribution of close homologs found in the *Proteobacteria*, Gram-positive *Actinobacteria*, *Bacteriodetes* and Archaeal *Methanobrevibacter* (Fig. S10). Within the *Shewanellae*, homologs of this quinol monooxygenase are both broadly distributed and tightly clustered in a highly conserved clade, with many homologs exhibiting 95–100% amino-acid-sequence identity. This may speak to the highly conserved function of this gene within the *Shewanellae* that is possibly distinct from the other homologs observed in other genera.

*SO_3660, SO_3662, and an electron uptake operon.* The AHDS$_{red}$ oxidation screen points to the existence of an electron-uptake operon in *S. oneidensis*. Disruption of any of the loci from *SO_3656* to *SO_3665* causes failure of AHDS$_{red}$ oxidation (Fig. 2g). This putative operon is putatively regulated by SO_3660, annotated as a transcriptional regulator. SO_3662 is annotated as an inner-membrane ferredoxin, supporting a direct role in electron transfer. In-frame deletions of the genes coding for SO_3660 and the putative inner-membrane-bound ferredoxin SO_3662 both disrupt electron uptake (Fig. 3c), but not deposition. Interestingly, the formal potentials quantified for each deletion mutant were statistically indistinguishable (Table S2).

Phylogenetic trees constructed from the 200 closest homologs of *SO_3600* and *SO_3662* revealed that these genes are highly conserved in *Gammaproteobacteria* and across numerous (≈100) *Shewanella* species (Fig. S11 and S12). *SO_3662* appears to be highly conserved among the order *Alteromodales* in particular.

**Role of electron uptake genes in *S. oneidensis*.** Like many other facultatively anaerobic microorganisms, including *E. coli*[33], *S. oneidensis* employs discrete anaerobic and aerobic electron-transport chains. Menaquinone is the dominant quinone used by *S. oneidensis* under anaerobic conditions where EET is used for mineral respiration[23]. Conversely, under aerobic conditions, ubiquinone is the dominant quinone used by *S. oneidensis*[34]. Furthermore, ubiquinone is important for reverse electron flow to NADH mediated by Complex I (a NADH:ubiquinone oxidoreductase) under cathodic conditions[13].

When taken together, the normally discrete machinery of aerobic and anaerobic electron-transport chains and the ability of *S. oneidensis* to couple reversal of the anaerobic EET pathway to $O_2$ reduction suggests that a specific connection between the two transport chains is likely to exist. However, to our knowledge, the mechanism allowing these organisms to transition between one electron transport chain, or quinone pool to another, is poorly understood.

We outline two possible mechanisms for a connection between the anaerobic EET pathway and the aerobic electron-transport chain in EEU in Fig. 1. First, the EET complex could transfer electrons to CymA, which then reduces menaquinone. Electrons could then be transferred from menaquinone to ubiquinone and into the aerobic electron-transport chain, finally arriving at Complex IV where they reduce $O_2$. This option seems intuitive as under anaerobic conditions using fumarate as an electron acceptor, *S. oneidensis* was shown to require menaquinone in addition to several components of the EET complex to uptake cathodic currents[10]. However, knockout of the gene coding for *cymA* does not disrupt cathodic electron uptake when using $O_2$ as a terminal electron acceptor[13].

The lack of involvement of CymA in cathodic electron uptake under aerobic conditions suggests a second option (Fig. 1): that cathodic electrons bypass the menaquinone pool under aerobic conditions.

We speculate that the putative quinol-interacting protein SO_0400 and the putative ferredoxin SO_3662 (and possibly proteins coded by nearby genes) are directly involved in connecting the reverse EET pathway during electron uptake and the aerobic electron-transport chain. Notably, the lack of a phenotype for most of these proteins under anodic conditions supports the hypothesis of a distinct connection between a subset of EET proteins and the aerobic electron-transport chain during EEU (Fig. 1). This work has also highlighted some genes potentially involved in cell signaling or transcriptional responses that may help aid in facilitating electron uptake under specific conditions (SO_0841, SO_0181 and SO_3660).

Though the motivation of this work stemmed from the application of *S. oneidensis* to electrosynthesis, it is likely that the process of EEU plays a role in *Shewanella* physiology and ecology. It has been shown that minerals in nature can serve a capacitive or electron-storing function for microbes[35]. While electron deposition and electron uptake from minerals such as magnetite were shown to function as both sinks and sources for different metabolisms, it is feasible that a single organism with both functionalities could utilize minerals in this way—functionally storing charge akin to a battery. Though iron oxidation has only been demonstrated in a single *Shewanella* strain[36], this could be due to the challenge of distinguishing biologic and abiotic iron

oxidation in the absence of growth. As *Shewanella* are not generally capable of carbon fixation, the process of EEU is unlikely to have evolved as a growth-linked metabolism. However, previous work has linked electron uptake to maintenance of cell biomass or decreasing the rate of cell death, which could suggest a role in allowing cells to conserve energy under nongrowth conditions[13]. Interestingly, these genes appear to be widely conserved across the *Shewanella*, as well as other marine *Gammaproteobacteria* (several of which have also been implicated in EEU). This supports the yet-unexplored ecological role for EEU in sediment and/or marine microbes, though our knowledge of the specific activity and role of this process is still at its inception.

## Conclusions

EEU holds significant potential for conversion of $CO_2$ and renewable electricity to complex organic molecules. While potential of unusual phenotypes like this can be limited by a lack of genetic understanding, especially in nonmodel organisms, synthetic biology can greatly expand the possibility for their improvement and application.

We used a whole-genome-knockout collection previously built with the rapid, low-cost knockout Sudoku method to screen the *S. oneidensis* genome for redox dye oxidation, a proxy for electron uptake. In this work, we have performed detailed electrochemical analyses, focusing on genes encoding proteins of unknown function. We have identified five previously uncharacterized genes in *S. oneidensis* that are involved in EEU from both solid phase and extracellular donors, coupled to both aerobic and anaerobic terminal electron acceptors.

Identification of these important genes lays the foundation for further genetic characterization of metal oxidation in nature, improvement of EEU in *S. oneidensis* and for synthetically engineering an electron-uptake pathway into easily engineered or synthetic microbes for powering recent advances in synthetic $CO_2$ fixation[15] and EET[37] in *E. coli*.

## Methods

**Genome-wide AHDS_red oxidation screen**. The *S. oneidensis* whole genome knockout collection[20,21] was screened for members unable to oxidize the redox dye anthra(hydro)quinone-2,6-disulfonate (AHDS_red for the reduced form and AQD-S_ox for the oxidized form)[23,24], and subsequently reduce either fumarate[22] or nitrate.

**Knockout collection construction**. The *S. oneidensis* whole-genome-knockout collection was previously built with the Knockout Sudoku whole-genome-knockout collection construction procedure[20,21]. Prior to high-throughput screening, the mutant collection was duplicated with a multiblot replicator (Catalog Number VP 407, V&P Scientific, San Diego CA, USA) into 96-well polypropylene plates containing 100 μL of LB with 30 μg mL$^{-1}$ kanamycin per well. The plates were labeled with barcodes and registration marks for identification in high-throughput analysis. The plates were sealed with an air-porous membrane (Aeraseal, Catalog Number BS-25, Excel Scientific) and grown to saturation overnight (at least 24 h) at 30 °C with shaking at 900 rpm.

**AHDS_red preparation**. Solutions of 25 mM AHDS_red for screening experiments were prepared electrochemically. About 200–1000-mL batches of 25 mM AQDS_ox were prepared by dissolving 10.307 mg of AQDS_ox powder (Catalog no. A0308, TCI America) per 1 mL of deionized water at 60 °C. The AQDS_ox solution was then transferred to a three-electrode electrochemical system (Catalog no. MF-1056, BASI Bulk Electrolysis Cell) inside a vinyl anaerobic chamber (97% $N_2$, 3% $H_2$ and <20 ppm $O_2$; Coy Laboratory Products, Grass Lake MI, USA). The system uses an Ag/AgCl reference electrode, a coiled Pt wire counterelectrode inside a fritted counterelectrode chamber and a reticulated vitreous carbon working electrode. The working-electrode potential was maintained at 700 mV vs. Ag/AgCl with a digitally controlled potentiostat (PalmSens, EmStat3). To enhance cell conductivity, 3 M $H_2SO_4$ acid was added to the working electrode, allowing the cell current to rise to ≈2 mA. AHDS_red reduction is assumed to be complete when the solution is translucent yellow in color, typically after ≈7 h. The pH of the AHDS_red stock solution was returned to 7.2 by addition of 3 M NaOH and the color changed from yellow to yellow-orange.

**Assay media preparation**. *Shewanella* basal media (SBM) was used for all AHDS_red oxidation assays. SBM consists of ammonium chloride ($NH_4Cl$) (0.0086 M), dibasic potassium phosphate ($K_2HPO_4$) (0.0013 M), monobasic potassium phosphate ($KH_2PO_4$) (0.0017 M), magnesium sulfate heptahydrate ($MgSO_4.7H_2O$) (0.0005 M), ammonium sulfate (($NH_4)_2SO_4$) (0.0017 M), and HEPES (0.1 M). The media was buffered to pH 7.2 with sodium hydroxide (NaOH) and sterilized by autoclave. Trace-mineral supplement (5 mL L$^{-1}$; Catalog no. MD-TMS, American Type Culture Collection (ATCC), Manassas, VA, USA) and vitamin supplement (5 mL L$^{-1}$; Catalog no. MD-VS, ATCC) were added aseptically. No carbon source (e.g., lactate) is added to the SBM to ensure that electrons flowing into the cell are only from AHDS_red.

Solutions of 25 mM potassium nitrate and 25 mM sodium fumarate adjusted to pH 7.2 and filter-sterilized were prepared as terminal electron acceptors for the AHDS_red oxidation screens. These solutions were transferred to an anaerobic chamber at least the evening prior to any experiment to allow deoxygenation.

**AHDS_red oxidation screens with nitrate**. Aliquots of 10 μL of culture of each mutant from the replicated *Shewanella* whole-genome-knockout collection were transferred to 96-well polystyrene assay plates filled with 50 μL of SBM using a 96-channel pipettor (Liquidator 96, Mettler-Toledo Rainin LLC, Oakland, CA, USA). The assay plates were transferred to an anaerobic chamber for de-oxygenation for at least 9 h. The cell activity was confirmed by transferring cells from the assay plate to LB after the 9 h resting period in the anaerobic chamber and observing growth.

Following deoxygenation, 40 μL of 1:1 mixture of 25 mM AHDS_red and 25 mM potassium nitrate were added to each well of the assay plates with a multichannel pipettor. The final concentration of both AHDS_red and potassium nitrate in each well was 5 mM. Each assay plate was immediately photographed after media addition with the macroscope. All plates were repeatedly imaged for at least ≈40 h. After completion of the experiment, photographs were analyzed with the macroscope image analysis software. Mixing instructions for a single well of a 96-well assay plate are shown in Table 1.

**AHDS_red oxidation screen with fumarate**. Aliquots of 1 μL of culture of each mutant from the replicated *S. oneidensis* whole-genome-knockout collection were transferred to 96-well polystyrene assay plates filled with 59 μL of SBM using a multiblot replicator (Catalog Number VP 407, V&P Scientific, San Diego CA, USA). The assay plates were transferred to an anaerobic chamber for deoxygenation for at least 9 h. The cell activity was confirmed after the resting period by subculturing a small amount of assay media in LB. AHDS_red oxidation was found to proceed much faster with fumarate than with nitrate, so the volume of cells added to the assay was reduced to make data collection more manageable.

Following deoxygenation, 40 μL of 1:1 mixture of 25 mM AHDS_red and 25 mM sodium fumarate were added to each well of the assay plates with a multichannel pipettor. The final concentration of both AHDS_red and sodium fumarate in each well was 5 mM. Each assay plate was immediately photographed after media addition with the macroscope. All plates were repeatedly imaged for at least ≈40 h. After completion of the experiment, photographs were analyzed with the

**Table 1 Solution mixing instructions for AHDS_red oxidation assay for each well in a 96-well plate.**

| AHDS_red Oxidation with Nitrate Assay | | | AHDS_red Oxidation with Fumarate Assay | | |
|---|---|---|---|---|---|
| **Component** | **Volume (μL)** | **Final Conc. (mM)** | **Component** | **Volume (μL)** | **Final Conc. (mM)** |
| *Shewanella* Basal Media | 50 | – | *Shewanella* Basal Media | 59 | – |
| Saturated Culture | 10 | – | Saturated Culture | 1 | – |
| 25 mM $KNO_3$ | 20 | 5 | 25 mM Na Fumarate | 20 | 5 |
| 25 mM AHDS_red | 20 | 5 | 25 mM AHDS_red | 20 | 5 |
| Total Volume (μL) | 100 | | Total Volume (μL) | 100 | |

macroscope image analysis software. Mixing instructions for a single well of a 96-well assay plate are shown in Table 1.

**Macroscope data acquisition system**. An automated photographic data-acquisition system was used to record the results of AHDS$_{red}$ oxidation assays. The device consists of a digital single-lens reflex (DSLR) camera (any member of the Canon Rebel Series) controlled by a macOS computer running a custom data-acquisition program that downloads images to the computer and timestamps them. The camera shutter is controlled by a single-switch foot pedal (vP-2, vPedal), leaving both hands free to manipulate 96-well plates. The camera is mounted to a frame constructed with extruded aluminum rails (T-slot). Barcodes attached to each plate enable images to be automatically sorted. Registration marks on the barcodes allow for identification of well positions and each well to be associated with a specific gene-knockout mutant. The device allows a stack of 200 96-well plates to be imaged in ≈15 min. This process can be repeated immediately, allowing each plate to be quasi-continuously imaged. A set of photographs of three generations of macroscope device, along with photographs taken with the macroscope, are shown in Figs. S13–S17.

This work has used two variants of the device: the first to image AHDS$_{red}$ oxidation in transparent 96-well assay plates, and the second to measure bacterial growth in 96-well storage plates covered with an air-porous membrane (Aeraseal, Catalog Number BS-25, Excel Scientific). In the first configuration, the camera is mounted above the AHDS$_{red}$ oxidation-assay plates. Each plate is placed inside a laser-cut acrylic holder and illuminated by an LED light pad from below (A920, Artograph). Barcodes are printed on transparent labels (Catalog no. 5660, Avery) and attached to the top of each plate. In the second configuration, the camera is mounted below the plate and illumination is provided from above with an LED light pad. A white barcode (SIDE-1000, Diversified Biotech, Dedham, MA, USA) is attached to the side of the plate and viewed through a 45° right-angle mirror (Catalog no. 47–307, Edmund Optics, Barrington, NJ, USA).

**Analysis of macroscope images**. A custom Macroscope Image Analyzer program was developed with Pyzbar[38], Pillow[39], Numpy[40], Matplotlib[41] and OpenCV-Python[42] using Python3. The program was used to process images taken with the macroscope device and aided in detection of loss-of-function mutants. The program handled the image analysis in four steps: creating barcodes, organizing the images collected with the macroscope by barcodes, collecting the data from the images, and finally presenting the data for analysis.

An additional image-analysis algorithm was developed with SciKit Image[43] and SimpleCV[44] to test images of 96-well storage plates for cross-contamination and growth-failure events by comparison with the collection catalog. The image-analysis algorithm updated the record for each well in the collection catalog with growth information to assist in rejection of false positives in the AHDS$_{red}$ oxidation screen due to growth failure.

The Macroscope Image Analyzer software is available on GitHub at https://github.com/barstowlab/macroscope-image-analyzer.

As AHDS$_{red}$ is oxidized to AQDS$_{ox}$, it changes color from yellow–orange to clear. Almost all information on the reduction state of the AHDS$_{red}$/AQDS$_{ox}$ dye can be found in the blue-color channel of the assay plate images. At the start of the assay, the intensity of the blue-color channel is low, and the dye is orange. As the AHDS$_{red}$/AQDS$_{ox}$ dye is oxidized and becomes clear, the intensity of the red channel remains approximately constant, with a small increase in green-channel intensity and a large increase in the blue-channel intensity. However, we found reporting the blue channel intensity as a proxy for the AHDS$_{red}$/AQDS$_{ox}$ redox state to be unintuitive.

To aid the reader and experimenter, we used the RGB color data,

$$\vec{C} = \begin{bmatrix} r \\ g \\ b \end{bmatrix}$$

to calculate a single number that represents how "yellow" a well is. The vector overlap (dot product) was calculated between the current color of the well and the most saturated yellow color in the assay photographic dataset.

The reference yellow color,

$$\vec{Y}_0 = \begin{bmatrix} 225 \\ 153 \\ 0 \end{bmatrix}$$

was calculated relative to the reference white color,

$$\vec{W}_0 = \begin{bmatrix} 255 \\ 255 \\ 255 \end{bmatrix}$$

Thus, the transformed yellow reference,

$$\vec{Y}_0' = \vec{Y}_0 - \vec{W}_0.$$

The transformed well color, relative to the white reference,

$$\vec{C}' = \vec{C} - \vec{W}_0.$$

The yellow intensity was calculated by normalizing the dot product between the transformed well color and the transformed yellow reference,

$$y = \frac{\vec{C}' \cdot \vec{Y}_0'}{|\vec{Y}_0'|^2}.$$

The normalized yellow intensity has a maximum value of 1 when the well is yellow and a minimum value of 0 when it is clear.

The time series of colors for each gene shown as colored circles above each gene in Fig. 2 were generated by an algorithm that interpolated the multireplicate average of mean well-center color values for that mutant at 0, 1, 2, 3 and 4 h after the initiation of the oxidation experiment. A visual explanation of this process is shown in Fig. S1. AHDS$_{red}$ oxidation rates reported in Fig. S5 were calculated by a linear fit to the linear portion of the yellow intensity curve with DATAGRAPH (Visual Data Tools).

**Bioelectrochemical cell construction and experimental conditions**. A 3-electrode electrochemical cell based on a design by Okamoto et al.[45] was assembled in-house, with the exception of a salt bridge that was included to contain the reference electrode (Part no. MF-2031, BASi, West Lafayette, IN, USA). As described, the cell consisted of a working electrode made of ITO- (indium tin oxide) plated glass (Delta Technologies, Ltd., Loveland, CO, USA), a counter-electrode of platinum wire and an Ag/AgCl reference electrode suspended in 1 M KCl (HCH Instruments, Inc., Austin, TX, USA). The reactor volume contained approximately 20 mL of liquid with a working-electrode surface area of 10.68 cm$^2$.

The electrochemical cell was used for chronoamperometry (CA) experiments that measure change in current over time and cyclic voltammetry (CV) experiments that measure current in response to a change in voltage. Both types of experiment were controlled with a 16-channel potentiostat (Biologic, France). In anodic CA experiments, the working electrode was poised at 422 mV vs. SHE. In cathodic CA experiments the working electrode was poised at −378 mV. This voltage minimizes hydrogen production while maximizing the biological cathodic activity. In CV experiments, the working-electrode potential was scanned between +422 mV and −378 mV vs. the SHE at a rate of 1 mV s$^{-1}$.

***Shewanella* culturing conditions**. Cultures of wild-type *S. oneidensis* and *S. oneidensis* mutants were grown from glycerol stocks overnight in Luria Broth (LB) prior to each experiment. Aerobic and anaerobic growth curves and electrochemical experiments for all strains were performed in a *Shewanella* defined media (SDM)[46], which we have found optimal for electrochemical experiments (note that SDM is not the same as SBM used in AHDS$_{red}$ oxidation assays). Aerobic cultures were performed in 50 mL volumes using 10 mM lactate as an electron acceptor shaking at 150 rpm at 30 °C. Kanamycin (Kan) and chloramphenicol (Chl) were added to LB and SDM media at concentrations of 100 µg mL$^{-1}$ and 34 µg mL$^{-1}$, respectively, for selection of transposon (Kan), clean gene deletion (Kan), and complementation (Chl + Kan) strains. The same growth conditions were used for anaerobic growth curves, with the exception that the media contained 20 mM fumarate and was purged with nitrogen gas for 10 min in serum vials.

For growth curves, strains were inoculated from overnight at a 100-fold dilution. Optical densities were recorded for triplicate cultures at 600 nm every 2–3 h. For cathodic growth, an overnight culture was backdiluted by a factor of 100 into SDM with 10 mM lactate and grown overnight. The overnight cultures grown in SDM were pelleted and resuspended in fresh SDM to an optical density at 600 nm of 0.1. About 20 mL of the resuspended culture was transferred to the working-electrode chamber of an electrochemical reactor. The reactor was attached to a 16-channel potentiostat (BioLogic) and the culture was anode-conditioned by poising the working electrode at +422 mV vs. SHE[12,13]. Anaerobic conditions needed to encourage biofilm formation and anodic current generation were maintained by continuous purging with N$_2$.

After approximately 24 h, the reactors were detached from the potentiostat and the media containing planktonic cells was carefully removed to avoid disturbing the biofilm on the working electrode. The reactor was then refilled with 20 mL of fresh carbon-free SDM[13]. The reactors were then reattached to the potentiostat and the working electrode was cathodically poised at −378 mV vs. SHE. Air was slowly bubbled into the reactors via an aquarium pump, until a steady stream was reached to provide a source of O$_2$.

To determine the portion of the cathodic current due to biological processes, the respiratory inhibitor Antimycin A was added to the electrochemical cell working-electrode chamber to a final concentration of 50 µM. Currents reported are the average difference between the steady-state currents pre and post Antimycin additions. To control for the effects of DMSO (the solvent for Antimycin A), blank injections of DMSO were made to the electrochemical cell and had no impact on current production. To confirm the effect of Antimycin A on biological current, addition of Antimycin A added to sterile minimal media in a reactor was performed and shown to have no impact on current production (Fig. S4).

**Protein collection and quantification**. Protein quantification was used to assess the total biomass in bioelectrochemical experiments. At the end of an electrochemical experiment, the spent media from the reactor (~20 mL) was collected and the biofilm was scraped from the working electrode. Biomass was centrifuged at $8000 \times g$ and the pellet was resuspended in 2 mL of 10% w/v NaOH. Total protein collected from the reactor was quantified using a Qubit TM Protein Assay Kit (Invitrogen, USA) according to the manufacturer's protocol.

**Biofilm imaging and cell counts**. Biofilms from $\delta SO\_0841$ and wild-type bioelectrochemical reactors were imaged on Nikon TI-E eclipse inverted microscope equipped with UV fluorescence. Cells were stained using the FM 4-64X lipid dye (Molecular Probes, Life Technologies Inc). At least 20 images were taken from each of three reactors for $\delta SO\_0841$ and wild-type replicate experiments. Cell dimensions and fluorescence intensity of a biofilm cross section were performed using the Nikon NIS-Elements software.

**Statistics and reproducibility**. Uncertainties in bioelectrochemical measurements were calculated by analysis of at least three replicates for each bioelectrochemical experiment. All statistical analyses were performed in Excel and/or R.

Cathodic midpoint potentials were calculated from cyclic voltammogram (CV) scans. A subsection of the CV scan (between the aforementioned parameters found under electrochemical conditions) was separated into a forward scan (from 222 mV to −322 mV vs. SHE) and a reverse scan (−322 mV to 222 mV). The scan range was chosen to contain the voltage at which maximum current production was achieved. Because linearity could not be assumed in these data to generate a function for a first-derivative analysis, an alternative method was used to analyze the data directly. A cubic smoothing spline (spar = 0.70) was applied to the current data in R to remove noise, but still captures the general trend. An approximate derivative was then taken from these values. From this approximate derivative, the maximum current produced, and its corresponding voltage potential were found for both the forward and reverse scans, which were then averaged to find the midpoint potential. The midpoint potential from each replicate was then pooled and averaged to obtain the reported value.

Anodic and cathodic currents were determined by chronoamperometry (CA). An average of the final 100 data points of each CA scan was taken to determine the average current achieved for each biological replicate. The replicates were then averaged to determine the average current produced for each strain. Biological current was determined by subtracting the average current post antimycin addition from the average cathodic current prior to addition. To determine if the average current of any of the mutants was significantly different from the wild-type, the cumulative current data were compiled in Excel and then fitted to a linear model in order to perform a type-II analysis of variance (ANOVA) test with R. Following the ANOVA test, anodic and cathodic currents for all mutants were compared with wild-type by Tukey's honestly significant difference (HSD) test to determine if any significant difference existed between them.

**Phylogenetic analysis**. Phylogenetic trees of relatives of the genes identified in this work were generated by search for homologs, homolog alignment and tree assembly by a maximum likelihood method. Approximately 120–200 homologs for each gene identified in this work were identified with the top homolog hit program that interrogates the Integrated Microbial Genomes and Microbiomes Database (https://img.jgi.doe.gov/)[47,48]. Homolog sets for each gene were aligned with the Muscle aligner[49] with default parameters.

Phylogenetic trees were generated for each homology set by a maximum likelihood method with RAxML 8.2.11[50]. In total, 100 trees were generated for each set of homologs for bootstrapping. Tree images and taxonomic metadata (Figs. S7–S11) were generated and annotated using the interactive Tree of Life (iTOL) program[51].

**Reporting summary**. Further information on research design is available in the Nature Research Reporting Summary linked to this article.

## Data availability
The datasets generated during and analyzed during the current study are available from the corresponding authors (A.R. and B.B.) on reasonable request. As the macroscope device for high-throughput image acquisition evolved organically, a blueprint does not exist. However, should a reader wish to construct one, B.B. and M.B. would be happy to consult.

## Code availability
The Macroscope Image Analyzer software is available at https://github.com/barstowlab/macroscope-image-analyzer and in ref. [52]. Unfortunately, due to licensing restrictions on the camera-control libraries, we cannot distribute the macroscope data-acquisition software online, but we can share our source code on request.

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

## Acknowledgements

We thank L. Jurgensen and S. Medin for programming assistance, B. Pian for experimental assistance, and A.M. Schmitz for review of this paper. This work was supported in the Barstow lab by a Career Award at the Scientific Interface from the Burroughs Welcome Fund, Princeton University startup funds, Cornell University startup funds and by U.S. Department of Energy Biological and Environmental Research grant DE-SC0020179. In the Rowe lab, this work was funded by Air Force Office of Scientific Research grant FA9550-19-1-0305 and University of Cincinnati startup funds. The Baym lab is supported by an award from the David and Lucille Packard Foundation.

## Author contributions

Conceptualization, A.R. and B.B.; Methodology, A.R., M.B. and B.B.; Investigation, A.R. F.S., L.T., J.S., O.A., I.A., L.K., M.B. and B.B.; Writing—Original Draft, A.R., L. T., F.S., and B.B.; Writing—Review and Editing, A.R., J.S., F.S. and B.B.; Funding Acquisition, A.R. and B.B.; Resources, A.R. and B.B.; Supervision, A.R. and B.B.; Data Curation, A.R., L.T., F.S. and B.B.; Visualization, A.R., F.S., L.T., J.S. and B.B. Formal Analysis, A.R., F.S., L.T., J.S. and B.B.

## Competing interests

The authors declare no competing interests.
