## [Peer Review File · Communications Biology]

Reviewers' Comments:

Reviewer #1:

Remarks to the Author:

The authors present a herculean effort to screen an entire knockout library of *S. oneidensis* for strains deficient in extracellular electron transfer from an external donor to the cell for fumarate, nitrate and oxygen reduction. The authors call this extracellular electron uptake (EEU). I found the study to be of high interest to both those who study electron uptake into bacterial cells from electrodes and also potentially for understanding how *S. oneidensis* may interact with available mineral or metal electron donors in the environment (although this was not tested here). The screening approach was interesting and yielded mutants that were also deficient in electron uptake from electrodes. The AHDS screening was a massive effort and portions were automated, which I thought was interesting and seems to be one of the first attempts of its size to evaluate so many mutants for EET in a high throughput manner. I was surprised the authors did not emphasize that more. Reading through the methods section it does seem that some specialized imaging and code were required to develop the screening approach and it might be worth mentioning more in the main text. I do wonder how easy it would be to reproduce a similar system although feel the method are adequately described. I also appreciate the statistical comparison of the CA. Overall, I enjoyed reading this article and although the authors could not conclusively determine the role of the putative electron uptake pathway, they made great progress to show the phenotype and open the door to further characterization.

How did using an anaerobic screening approach affect the outcome of the screen? It's not clear to me why mutants were screened with nitrate and fumarate but then switched to O₂ for electrode assays. Why not just stay with fumarate and nitrate?

I may have missed this somewhere in the main text but the methods say (line 421) that the biofilms had to be anode conditioned before switching to cathodes. How did this work for mutant strain SO_0841 that had both an anode and cathode phenotype? Is it possible that other strains were affected during anodic growth and therefore didn't have a cathode phenotype? Like SO_4412?

I like that the authors compared the total protein from each reactor to assess whether the mutant strains had an impact on biofilm growth. This is important because current could be impacted by number of cells on the electrode. Did the authors attempt and confocal microscopy of the electrode biofilms to look at viability or biofilm morphology?

Figure 1 – Are the authors proposing that "Possible mechanism 2..." is composed of SO_0400 and SO_3663? The authors have drawn a sort of "walkway" rather than protein boxes but it is not labeled so I'm inferring from the main text. This should be clarified.

Figure 3 – This figure is a little confusing regarding SO_3662. Figure 3A is called "5 Novel Gene Disruption..." but does not show SO_3662, presumably because it was not tested in this initial assay as the main text states. If that is the case, the heading should be changed.

Minor points

It is the authors' discretion to create terminology but some of the phrases are a little awkward for me to wrap my mind around. EEU is ok but generally EET refers to both forward and reverse so creating EEU makes it sound like EET is only going out of the cell. Also the phrase "electron deposition" when describing anodic current production is a little unusual.

Line 47 – do you mean electroautotrophic not electroactive?

Line 72 – can you clarify here if AQDS is extracellular only? I am inferring that it is but if it goes into the periplasm then that would surely affect the results.

Line 289 – here it says reduced AHDS is translucent red but line 360 describes it as yellow-orange

Line 409 – Are SDM and SBM the same medium?

Line 427 – "cathodically poised at -378 mV SHE"?? I thought it was -278 mV SHE? Also, how was this

potential selected?

Line 447 – different potential window given here for CV than that given on line 404.

Reviewer #2:

Remarks to the Author:

Summary: This manuscript describes a high-throughput screen of an existing *Shewanella oneidensis* MR-1 transposon mutant library for genes essential for extracellular electron uptake (EEU). 150 of the mutants demonstrated a reduction in EEU ability, of which 109 could be explained by existing knowledge of *Shewanella* metabolism, respiration, transport, etc. A subset of the 41 unexplained mutants was further tested for capacity for extracellular electron transfer (EET; under both anodic and cathodic potential) in a bioelectrochemical system. This led to identification of 5 gene disruptions that caused a significant decrease in EEU but no change in electron transfer to an anode. Based on predicted structure and homology to other organisms, two of the genes are theorized to be involved in extracellular electron uptake only, not electron deposition, during aerobic respiration. Specifically, they are theorized to be a part of an electron transfer system that bypasses CymA and menaquinones, going directly to ubiquinone reductases. The remaining 3 genes were predicted to encode signaling or transcription control proteins involved in EET. This screening process was a rapid and effective means for hypothesis generation in the area of *Shewanella* EEU. While a specific mechanism for EEU was not fully elucidated in this study, this is an important report and dataset and provides interesting hypotheses for the field going forward.

Major comments:

1. In lines 80-90 and figure 2, it is not clear how the results from assays with AHDS and nitrate were used versus assays with AHDS and fumarate. Was the entire library tested in both versions of the assay? If so, were the 150 genes those that showed reduction in both conditions or only one or the other? While these questions can mostly be answered using the supplementary information, the information should be also be provided in main text and the figure 2 legend.

2. Figure 2

a. For panels D-G, are these results with nitrate or fumarate as the electron acceptor? Also, it appears that the heading for these panels is incorrect and should say that these robustly disrupt e- uptake from AHDS, rather than the electrode.

3. Figure 3

a. The heading for Figure 3A indicates that 5 mutants in genes previously not known to be involved in EET showed significant reductions. However, out of the 6 marked as significant, 2 are previously known (*mtrA* and *mtrC*). Doesn't that mean there are 4 novel genes shown in 3A?

b. Similarly, it seems like the heading for 3B should be 3 out of 4 instead of 4 out of 5.

c. What is the significance of the arrows marking $\delta mtrA$ in 3A and 3B? If denoting a control strain, shouldn't $\delta mtrC$ be marked also?

4. In line 123-127, it is mentioned that δSO_{3662} was not initially included in the cathode tests, but it is not explained what led to its later inclusion. I'm assuming it is something to do to the proximity to δSO_{3660} , but what led to its inclusion over other genes in this operon?

5. In line 156-158, it is stated that *SO_0841* homologs are found in other electroactive bacteria, thus supporting a conserved role of this gene in electron uptake, but also mentioned that homologs are found across a range of proteobacteria. I am not sure I agree with the idea that this gene has a conserved role in electron uptake if it is found in other non-electroactive bacteria. If this is the case, please expand more on what led to this assertion.

Minor comments:

Line 129-130: states that the deletion mutant phenotypes cannot be explained by changes to biofilm or growth rate, then references Table S3. If the protein assay is used as the biofilm measurement, this should also reference Table S2.

Line 154: If using the protein quantification as a biofilm measurement, I believe it should be Table S2.

Line 205: states that *S. oneidensis* is a facultative microorganism. Do you mean facultative anaerobe?

Line 416: change 'ever' to 'every'

Reviewer #3:

Remarks to the Author:

Shewanella oneidensis has long been considered a model organism for microbial extracellular electron transfer, especially for the donation of electrons to extracellular electron acceptors. It also has been observed early on that *S. oneidensis* can reverse this function and accept electrons from extracellular electron donors. This well designed study tries to elucidate the functional differences in electron discharge and electron uptake and to identify essential components of electron uptake pathways. A knock-out library of all non-essential genes was combined with a colorimetric assay from extracellular oxidation capacity. Many relevant genes involved in expected biochemical processes were identified, but also some genes (41) with unknown function proved essential for electron uptake. A selection of these genes were subsequently tested for electron uptake from a cathode in a bioelectrochemical system and for 5 unknown gene knockdowns the capacity for cathodic electron uptake was significantly impaired. With a *in silico* analysis and phylogenetic comparison of these genes, a first detailed pathway model for electron uptake was developed.

The paper is very well written and the many complex and detailed datasets (suppl. Information) have been nicely condensed with statistical methods to report the significant results. I have no major concerns with this manuscript. However, I have the following suggestions/questions, which might help improve the manuscript further (in order of appearance):

- L74 "Simplified AHDS..." – This sentence can be deleted from the introduction. This already points to results, which are introduced later on.
- L77 While it is good to highlight the experimental approach in the introduction, the reference to Fig 3 should be removed here. You are going too far ahead.
- Methods: somehow it does not really become clear if the assay medium contained a carbon source (lactate)? Was lactate used for all experiments? Please clearly state. If it was not used for the assay, did you check the activity of the cells after the 9 hour resting period in the anaerobic chamber? Did this affect the cells?
- L430-432: Please state more clearly if then only the biological part of the cathodic activity (inhibited part) was used for the current comparisons?
- Figure 3: please explain the blue shaded areas in the caption.
- Figure S6: It is not quite clear to me what "aerobic to anaerobic" and "anaerobic to aerobic transitional growth curves" means? Just aerobic and anaerobic growth curves? Please explain better.

Buz Barstow
Assistant Professor
Department of Biological and Environmental Engineering
Cornell University

228 Riley Robb Hall
111 Wing Drive
Ithaca, NY 14853
T: (607) 351-7356
E: bmb35@cornell.edu
W: barstow.bee.cornell.edu

24th May 2021

Text from the reviews are quoted in *italics*. The location of changes to the manuscript are noted in responses to specific comments by the reviewers and are highlighted in red or blue (two colors are used to distinguish separate consecutive changes) in the manuscript itself along with a tag that denotes the reviewer comment this change addresses (e.g., **RC1.1**).

To help track the reviewer comments and corresponding responses, we have summarized the changes and their location in the revised manuscript in a table attached to the end of this letter. Alternatively, to find all changes associated with a reviewer comment, search in the main text and supplementary information for these tags (e.g., **RC1.1**).

Response to Reviewer 1

In summary **Reviewer 1** comments “[t]he authors present a herculean effort to screen an entire knockout library of *S. oneidensis* for strains deficient in extracellular electron transfer from an external donor”.

RC1.1 *“The AHDS screening was a massive effort and portions were automated, which I thought was interesting and seems to be one of the first attempts of its size to evaluate so many mutants for EET in a high throughput manner. I was surprised the authors did not emphasize that more. Reading through the methods section it does seem that some specialized imaging and code were required to develop the screening approach and it might be worth mentioning more in the main text. I do wonder how easy it would be to reproduce a similar system although feel the method are adequately described”.*

We really appreciate this comment from **Reviewer 1**. Screening the *Shewanella oneidensis* whole genome knockout collection was an enormous amount of work (8 years, by two labs), but it isn’t our style to play this up too much. We feel the article might turn into a bit of a misery memoir if we did!

Additionally, a major portion of this breakthrough came about in two earlier works. First we (Barstow Lab) had to create the Knockout Sudoku technology for building knockout collections rapidly at ultra low cost, and then apply it to the construction of the *S. oneidensis* knockout collection [Baym2016a]. This collection dramatically reduced our workload in screening the whole *S. oneidensis* genome. Second, we (Rowe Lab) had to develop techniques for measuring electron uptake from *S. oneidensis* biofilms and demonstrating regeneration of ATP and NADH [Rowe2018a].

The high-throughput screening technology did require development of a specialized image acquisition hardware, image acquisition software, and analysis software (the macroscope system). The macroscope system was first described in our Knockout Sudoku article [Baym2016a] where it was used to functionally validate the *S. oneidensis* knockout collection by screening for mutants deficient in extracellular electron transfer to AQDS (the well known electron deposition phenotype, [Newman2000a, Shyu2002a]).

There are two major reasons that prevent us from sharing more about the macroscope. First, the macroscope hardware evolved organically. As a result, no blueprints for it exist, although it is easy to construct one from extruded aluminum (T-slot) pieces to suit any particular application. In fact, we would advise that a prospective user design their own macroscope to suit their needs, rather than relying on our exact design. Multiple versions of the macroscope system exist (both hardware and software), some of

which were used in this work, each with different capabilities. For the curious reader, we have included some additional photographs of the macroscope used to collect data for this study in new **Figs. S13 to S17**.

Secondly, the image acquisition software that runs the macroscope was written with camera control libraries provided by Canon. Due to licensing restrictions, we are unable to freely distribute this software at this time. However, we did provide a link to a Github repository containing the image analysis code used in this work. We have included additional statements in the **Code Availability** section. We have also included an additional statement in the revised main text on indicating our willingness to assist anyone who wishes to build a macroscope.

RC1.2 *“How did using an anaerobic screening approach affect the outcome of the screen? It’s not clear to me why mutants were screened with nitrate and fumarate but then switched to O₂ for electrode assays. Why not just stay with fumarate and nitrate?”*

This is a great question that we address in the revised main text. Initially, we really wanted to come up with a high-throughput assay that worked with O₂, as its very high redox potential (high electronegativity) and high availability allows the greatest opportunity for generating proton motive force, and hence storing energy and regenerating ATP and NADH. Both of these factors are incredibly important for electromicrobial production: turning CO₂ and renewable electricity into important chemicals. We tried to make this assay work for several weeks, but could not come up with a reliable format. AHDS_{red} is exquisitely sensitive to oxidation by O₂, meaning that we needed to use a terminal electron acceptor that could not directly oxidize AHDS_{red} but instead has to rely on *Shewanella* as an intermediate.

Secondly, we were most interested in parts of the electron uptake pathway that were not responsible for terminal electron acceptor reduction. As a result, using different terminal electron acceptor allowed us to exclude genes that would only affect electron uptake with a single terminal electron acceptor.

RC1.3 *“I may have missed this somewhere in the main text but the methods say (line 421) that the biofilms had to be anode conditioned before switching to cathodes. How did this work for mutant strain SO_0841 that had both an anode and cathode phenotype? Is it possible that other strains were affected during anodic growth and therefore didn’t have a cathode phenotype? Like SO_4412?”*

We thank the reviewer for this comment. In our previous work, we screened mutants of the known EET pathway (known anodic phenotypes) for cathodic phenotypes, and we were concerned with the potential for biofilm attachment deficiencies to be driving the effects we were observing (i.e., loss of cathodic current). As such we imaged all the biofilms, performed cell counts and quantified total protein from the electrode biofilms [Rowe2018a]. In that work we observed no evidence of differences in biofilm formation from the mutants we screened compared to wild-type. Notably, though the variance was high for the protein measurements, they were no less variable than the microscopy count data and so we adopted use of protein quantification as a way to measure biofilm/cell abundance in our current work. Given that work presented here tests new genes of unknown function, we used protein quantification to ensure we were not observing cathode phenotypes that are driven by biofilm formation deficiencies. Our protein data did not demonstrate any such trends. So while it is perhaps surprising that genes like *SO_0841* which has an anodic phenotype, has similar biofilm growth and so this is not likely driving the cathode phenotype, this is a very similar case to what we see in genes like *mtrA* and *mtrC*. As a counterpoint, some genes that have strong anode phenotypes but lack significant cathode phenotypes. For example *cymA*, and *SO_4412* which supports successful biofilm formation that yields cathodic activity even under conditions where anode activity is reduced. We have added clarifying text that to the revised manuscript that we hope addresses this concern.

RC1.4 *“I like that the authors compared the total protein from each reactor to assess whether the mutant strains had an impact on biofilm growth. This is important because current could be impacted by number of cells on the electrode. Did the authors attempt and confocal microscopy of the electrode biofilms to look at viability or biofilm morphology?”*

As mentioned, we did this in our previous work and did not notice any trends in terms of cell numbers. However, we did not perform microscopy as part of this work. To alleviate the concern about biofilm morphology in, we acquired and analyzed images of δSO_{0841} biofilms as this was from a previously un-investigated strain and has the most relevant phenotype being deficient in both anodic and cathodic current. We compare these biofilms to wild-type in terms of percent electrode coverage (assessed by fluorescence intensity) and cell morphology. We have added additional comments to the main text, an additional supplementary figure (new **Fig. S7**), and an additional methods subsection (**Biofilm Imaging and Cell Counts**) to address this point. No noticeable difference in biofilm morphology or cell size was noted. In terms of viability, previous work in *Shewanella* biofilm ecology has demonstrated that attachment is an energetically active process, requiring ATP and/or an electrochemical gradient or significant dissolution occurs [Saville2011a]. This is distinct from other biofilm forming species like *Pseudomonas* where non-viable cells remain attached. We did test cell morphology by microscopy.

RC1.5 *“Figure 1 – Are the authors proposing that “Possible mechanism 2...” is compose of SO_0400 and SO_3663? The authors have drawn a sort of “walkway” rather than protein boxes but it is not labeled so I’m inferring from the main text. This should be clarified.”*

Thank you for pointing this out. We have added clarifying text to the figure caption.

RC1.6 *“Figure 3 – This figure is a little confusing regarding SO_3662. Figure 3A is called “5 Novel Gene Disruption...” but does not show SO_3662, presumably because it was not tested in this initial assay as the main text states. If that is the case, the heading should be changed.”*

We have modified the heading in **Fig. 3** and add clarifying text the caption. This response also addresses concerns in **RC2.4**.

RC1.7 *“It is the authors’ discretion to create terminology but some of the phrases are a little awkward for me to wrap my mind around. EEU is ok but generally EET refers to both forward and reverse so creating EEU makes it sound like EET is only going out of the cell. Also the phrase “electron deposition” when describing anodic current production is a little unusual.”*

We very much appreciate this comment, and sympathize with this point of view. However, we felt that the option of creating new terminology to that electron uptake is distinct from electron deposition) was the least worst option. On the phrase “electron deposition”, we also considered the phrase “electron outflow” but considered this even more clumsy.

RC1.8 *“Line 47 – do you mean electroautotrophic not electroactive?”*

Yes, we do mean electroautotrophic, and have amended the main text.

RC1.9 *“Line 72 – can you clarify here if AQDS is extracellular only? I am inferring that it is but if it goes into the periplasm then that would surely affect the results.”*

This is a very astute observation. Our best guess, based upon results by Shyu *et al.* [Shyu2002a], is that AHDS_{red}/AQDS_{ox} can enter the cell, but is rapidly pumped out by a toxin efflux pump. Thus, we suspect that the cell maintains a much lower interior concentration of AHDS_{red}/AQDS_{ox} than is found outside the cell. We found in an earlier work [Baym2016a] that knocking out cell surface proteins like the Mtr EET complex does reduce AQDS_{ox} reduction rate, suggesting these proteins have the opportunity to transfer electrons to AHDS_{red}/AQDS_{ox}. AHDS_{red}/AQDS_{ox} is not a perfect proxy for EET, but is far better than nothing. You could say we are more than glass-half-full about it. We have added clarifying remarks to the main text.

RC1.10 “Line 289 – here it says reduced AHDS is translucent red but line 360 describes it as yellow-orange.”

We have amended the main text to make our color descriptions consistent.

RC1.11 “Line 409 – Are SDM and SBM the same medium? Line 409 – Are SDM and SBM the same medium?”

This is a good catch. No, SBM and SDM are not the same thing. We have added clarifying remarks to revised main text.

RC1.12 “Line 427 “–cathodically poised at -378 mV SHE”?? I thought it was -278 mV SHE? Also, how was this potential selected?”

We have corrected the text to show that the cathode was poised -378 mV to minimize H₂ production while maximizing current generation. We have added clarifying remarks to the main text.

RC1.13 “Line 447 – different potential window given here for CV than that given on line 404.”

We clarify in the revision that the first derivative analysis was run on a subsection of the full CV scan (222 to -322 mV). This was to focus on capturing the peak analysis region for the *Shewanella* electron uptake feature.

Response to Reviewer 2

In summary **Reviewer 2** notes “While a specific mechanism for EEU was not fully elucidated in this study, this is an important report and dataset and provides interesting hypotheses for the field going forward.”

Reviewer 2 makes 12 specific recommendations for revisions that we address here and in the revised manuscript.

RC2.1 “In lines 80-90 and figure 2, it is not clear how the results from assays with AHDS and nitrate were used versus assays with AHDS and fumarate. Was the entire library tested in both versions of the assay? If so, were the 150 genes those that showed reduction in both conditions or only one or the other? While these questions can mostly be answered using the supplementary information, the information should be also be provided in main text and the figure 2 legend.”

This is an astute point. First, we have reduced the number of claimed hits found by our assay to 149, as one (*δpepD*) was insufficiently robust. The 149 hits found by our assay caused AHDS_{red} oxidation failure

when using nitrate or fumarate (and in many cases both) as a terminal acceptor. We have added clarifying remarks in the revised main text.

RC2.2 *“Figure 2. For panels D-G, are these results with nitrate or fumarate as the electron acceptor?”*

All time courses use fumarate. We have added clarifying text to the revised manuscript.

RC2.3 *“Figure 2. Also, it appears that the heading for these panels is incorrect and should say that these robustly disrupt e- uptake from AHDS, rather than the electrode.”*

Actually, we really do mean an electrode. All of the 149 mutants we mention in this figure disrupt AHDS_{red} oxidation, but the 5 highlighted in panels **D** to **G** are novel and were found to robustly disrupt e⁻ uptake from a cathode. We have added clarifying text to the caption in the revised main text.

RC2.4 *“Figure 3. The heading for Figure 3A indicates that 5 mutants in genes previously not known to be involved in EET showed significant reductions. However, out of the 6 marked as significant, 2 are previously known (mtrA and mtrC). Doesn't that mean there are 4 novel genes shown in 3A?”*

See response to **RC1.6**.

RC2.5 *“Figure 3. Similarly, it seems like the heading for 3B should be 3 out of 4 instead of 4 out of 5.”*

We have clarified this in the revised main text.

RC2.6 *“Figure 3. What is the significance of the arrows marking $\delta mtrA$ in 3A and 3B? If denoting a control strain, shouldn't $\delta mtrC$ be marked also?”*

The arrows indicated that these knockout mutants significantly disrupted electron uptake from the cathode, but the box width was too small to display color, so an arrow was used. We have clarified this with a revised caption.

RC2.7 *“In line 123-127, it is mentioned that δSO_{3662} was not initially included in the cathode tests, but it is not explained what led to its later inclusion. I'm assuming it is something to do to the proximity to δSO_{3660} , but what led to its inclusion over other genes in this operon?”*

Basically, we did not discover the involvement of genes surround *SO_3660* and *SO_3662* until much later in the gene discovery process (after we had sent many mutants for electrochemical testing). Simply put, *SO_3662* had been on our radar for years, the other operon members had not been. We have added some clarifying text to the revised manuscript to describe the gene discovery process. We will likely get to the other putative operon members in a future publication.

RC2.8 *“In line 156-158, its stated that *SO_0841* homologs are found in other electroactive bacteria, thus supporting a conserved role of this gene in electron uptake, but also mentioned that homologs are found across a range of proteobacteria. I am not sure I agree with the idea that this gene has a conserved role in electron uptake if it is found in other non-electroactive bacteria. If this is the case, please expand more on what led to this assertion.”*

This is a very astute point. As electron uptake is not a widely tested physiology in microbes, it has only been described in *Shewanella* for example within the last 10 years, it is hard to draw too many conclusions from the wide distribution of this gene within the *Proteobacteria*. However, we believe it is notable that homologs are found in strains that have been specifically characterized for electron uptake on electrodes. We have re-written this section to clarify this point and softened the language on predicting a role for this gene in electron uptake in the revised main text.

RC2.9 “Line 129-130: states that the deletion mutant phenotypes cannot be explained by changes to biofilm or growth rate, then references Table S3. If the protein assay is used as the biofilm measurement, this should also reference Table S2.”

We have added references to **Tables S2** and **S3**.

RC2.10 “Line 154: If using the protein quantification as a biofilm measurement, I believe it should be Table S2.”

We have corrected the table reference (to **Table S2**) in the revised main text.

RC2.11 “Line 205: states that *S. oneidensis* is a facultative microorganism. Do you mean facultative anaerobe?”

Yes, we have revised the main text.

RC2.12 “Line 416: change ‘ever’ to ‘every’”

We have amended the text in the revised manuscript.

Response to Reviewer 3

Reviewer 3 notes “[t]his well designed study tries to elucidate the functional differences in electron discharge and electron uptake and to identify essential components of electron uptake pathways” and that “[t]he paper is very well written and the many complex and detailed datasets (suppl. Information) have been nicely condensed with statistical methods to report the significant results. I have no major concerns with this manuscript”.

Reviewer 3 makes 6 specific recommendations for revisions to our manuscript.

RC3.1 “L74 “Simplified AHDS...” – This sentence can be deleted from the introduction. This already points to results, which are introduced later on.”

This sentence has been deleted.

RC3.2 “L77 While it is good to highlight the experimental approach in the introduction, the reference to Fig 3 should be removed here. You are going too far ahead.”

We have amended the main text.

RC3.3 *“Methods: somehow it does not really become clear if the assay medium contained a carbon source (lactate)? Was lactate used for all experiments? Please clearly state. If it was not used for the assay, did you check the activity of the cells after the 9 hour resting period in the anaerobic chamber? Did this affect the cells?”*

We have added clarifying remarks regarding carbon source in the assay medium (there is none), and cell activity after resting in the anaerobic chamber.

RC3.4 *“L430-432: Please state more clearly if then only the biological part of the cathodic activity (inhibited part) was used for the current comparisons?”*

The following has been added to the methods section: “Currents reported are the average difference between the steady state currents pre and post Antimycin additions”

RC3.5 *“Figure 3: please explain the blue shaded areas in the caption.”*

These regions are meant to depict the wild type mean and standard deviation range for cathodic and anodic current respectively. Better description of this has been added to the **Fig. 3** legend.

RC3.6 *“Figure S6: It is not quite clear to me what “aerobic to anaerobic” and “anaerobic to aerobic transitional growth curves” means? Just aerobic and anaerobic growth curves? Please explain better.”*

The figure legend for **Fig. S6** has been clarified, by explaining the difference in pre-growth vs. growth conditions for these curves.

We would like to thank the reviewers for their comments which have allowed us to improve this article. We hope the revised manuscript is acceptable.

Yours sincerely,

Buz Barstow and Annette Rowe (for the authors)

References

- [Baym2016] Baym, M., Shaket, L., Anzai, I. A., Adesina, O. & Barstow, B. Rapid construction of a whole-genome transposon insertion collection for *Shewanella oneidensis* by Knockout Sudoku. *Nat Commun* **7**, 13270 (2016).
- [Rowe2018a] Rowe, A. R. *et al.* Tracking Electron Uptake from a Cathode into *Shewanella* Cells: Implications for Energy Acquisition from Solid-Substrate Electron Donors. *Mbio* **9**, e02203-17 (2018).
- [Saville2011a] Saville, R. M., Rakshe, S., Haagensen, J. A. J., Shukla, S. & Spormann, A. M. Energy-Dependent Stability of *Shewanella oneidensis* MR-1 Biofilms. *J Bacteriol* **193**, 3257–3264 (2011).
- [Shyu2002a] Shyu, J. B. H., Lies, D. P. & Newman, D. K. Protective Role of *tolC* in Efflux of the Electron Shuttle Anthraquinone-2,6-Disulfonate. *J Bacteriol* **184**, 1806–1810 (2002).

Response Summary

Reviewer Comment Number	Comment	Response Summary	Response Start Location(s)
1.1	The AHDS screening was a massive effort and portions were automated, which I thought was interesting and seems to be one of the first attempts of its size to evaluate so many mutants for EET in a high throughput manner. I was surprised the authors did not emphasize that more. Reading through the methods section it does seem that some specialized imaging and code were required to develop the screening approach and it might be worth mentioning more in the main text. I do wonder how easy it would be to reproduce a similar system although feel the method are adequately described	Revised main text New Figs. S13 to S17	Main Text Lines 375, 534, 538 SI Text Lines 46 Figs. S13 to S17
1.2	How did using an anaerobic screening approach affect the outcome of the screen? It's not clear to me why mutants were screened with nitrate and fumarate but then switched to O ₂ for electrode assays. Why not just stay with fumarate and nitrate?	Revised main text	Main Text Lines 85, 107
1.3	I may have missed this somewhere in the main text but the methods say (line 421) that the biofilms had to be anode conditioned before switching to cathodes. How did this work for mutant strain SO_0841 that had both an anode and cathode phenotype? Is it possible that other strains were affected during anodic growth and therefore didn't have a cathode phenotype? Like SO_4412?	Added clarifying remarks to revised main text.	Main Text Lines 151, 153
1.4	I like that the authors compared the total protein from each reactor to assess whether the mutant strains had an impact on biofilm growth. This is important because current could be impacted by number of cells on the electrode. Did the authors attempt and confocal microscopy of the electrode biofilms to look at viability or biofilm morphology?	New Fig. S7 Additional Methods subsection Revised main text	Main Text Lines 487 Fig. S7
1.5	Figure 1 – Are the authors proposing that “Possible mechanism 2...” is compose of SO_0400 and SO_3663? The authors have drawn a sort of “walkway” rather than protein boxes but it is not labeled so I'm inferring from the main text. This should be clarified.	Added clarifying remarks in the main text.	Main Text Lines 570
1.6	Figure 3 – This figure is a little confusing regarding SO_3662. Figure 3A is called “5 Novel Gene Disruption...” but does not show SO_3662, presumably because it was not tested in this initial assay as the main text states. If that is the case, the heading should be changed.	Modified headings in Fig. 3 and added clarifying text to caption.	Main Text Lines 597, 598, 607, 613
1.7	It is the authors' discretion to create terminology but some of the phrases are a little awkward for me to wrap my mind around. EEU is ok but generally EET refers to both forward and reverse so creating EEU makes it sound like EET is only going out of the cell. Also the phrase “electron deposition” when describing anodic current production is a little unusual.	See response in letter to editor	
1.8	Line 47 – do you mean electroautotrophic not electroactive?	Changed text to electroautotrophic	Main Text Line 47

Reviewer Comment Number	Comment	Response Summary	Response Start Location(s)
1.9	Line 72 – can you clarify here if AQDS is extracellular only? I am inferring that it is but if it goes into the periplasm then that would surely affect the results.	Added clarifying remarks to revised main text.	Main Text Line 74
1.10	Line 289 – here it says reduced AHDS is translucent red but line 360 describes it as yellow-orange.	Amended main text	Main Text Line 324, 325
1.11	Line 409 – Are SDM and SBM the same medium? Line 409 – Are SDM and SBM the same medium?	Added clarifying remarks to revised main text.	Main Text Line 449
1.12	Line 427 – “cathodically poised at -378 mV SHE”?? I thought it was -278 mV SHE? Also, how was this potential selected?	Corrected voltage and added clarifying remarks to revised main text	Main Text Line 443
1.13	Line 447 – different potential window given here for CV than that given on line 404.	Added clarifying text to revised main text	Main Text Line 497
2.1	In lines 80-90 and figure 2, it is not clear how the results from assays with AHDS and nitrate were used versus assays with AHDS and fumarate. Was the entire library tested in both versions of the assay? If so, were the 150 genes those that showed reduction in both conditions or only one or the other? While these questions can mostly be answered using the supplementary information, the information should be also be provided in main text and the figure 2 legend.	Added clarifying remarks in the main text.	Main Text Line 90, 575, 577
2.2	Figure 2. For panels D-G, are these results with nitrate or fumarate as the electron acceptor?	Added clarifying text to revised main text.	Main Text Line 593
2.3	Figure 2. Also, it appears that the heading for these panels is incorrect and should say that these robustly disrupt e- uptake from AHDS, rather than the electrode.	Added clarifying text to revised main text.	Main Text Line 589
2.4	Figure 3. The heading for Figure 3A indicates that 5 mutants in genes previously not known to be involved in EET showed significant reductions. However, out of the 6 marked as significant, 2 are previously known (mtrA and mtrC). Doesn't that mean there are 4 novel genes shown in 3A?	See response to RC1.6	
2.5	Figure 3. Similarly, it seems like the heading for 3B should be 3 out of 4 instead of 4 out of 5.	See response to RC1.6	
2.6	Figure 3. What is the significance of the arrows marking δ mtrA in 3A and 3B? If denoting a control strain, shouldn't δ mtrC be marked also?	Revised figure caption in revised main text	Main Text Line 604
2.7	In line 123-127, it is mentioned that δ SO_3662 was not initially included in the cathode tests, but it is not explained what led to its later inclusion. I'm assuming it is something to do to the proximity to δ SO_3660, but what led to its inclusion over other genes in this operon?	Added clarifying remarks in revised main text. In addition, see responses to RC1.6 , RC2.4 and RC2.5 .	Main Text Line 136, 146

Reviewer Comment Number	Comment	Response Summary	Response Start Location(s)
2.8	In line 156-158, its stated that SO_0841 homologs are found in other electroactive bacteria, thus supporting a conserved role of this gene in electron uptake, but also mentioned that homologs are found across a range of proteobacteria. I am not sure I agree with the idea that this gene has a conserved role in electron uptake if it is found in other non-electroactive bacteria. If this is the case, please expand more on what led to this assertion.	We caveated this statement in the revised main text	Main Text Line 189
2.9	Line 129-130: states that the deletion mutant phenotypes cannot be explained by changes to biofilm or growth rate, then references Table S3. If the protein assay is used as the biofilm measurement, this should also reference Table S2.	Added revised references in revised main text	Main Text Line 153
2.10	Line 154: If using the protein quantification as a biofilm measurement, I believe it should be Table S2.	Amended table reference and added new supplementary figure	Main Text Line 186
2.11	Line 205: states that S. oneidensis is a facultative microorganism. Do you mean facultative anaerobe?	Amended text in revised manuscript	Main Text Line 240
2.12	Line 416: change ‘ever’ to ‘every’	Amended text in revised manuscript	Main Text Line 459
3.1	L74 “Simplified AHDS...” – This sentence can be deleted from the introduction. This already points to results, which are introduced later on.	This sentence has been deleted	Main Text Line 79
3.2	L77 While it is good to highlight the experimental approach in the introduction, the reference to Fig 3 should be removed here. You are going too far ahead.	Reference removed	Main Text Line 81
3.3	Methods: somehow it does not really become clear if the assay medium contained a carbon source (lactate)? Was lactate used for all experiments? Please clearly state. If it was not used for the assay, did you check the activity of the cells after the 9 hour resting period in the anaerobic chamber? Did this affect the cells?	Added clarifying remarks to revised main text.	Main Text Lines 333 and 342
3.4	L430-432: Please state more clearly if then only the biological part of the cathodic activity (inhibited part) was used for the current comparisons?	Added clarifying remarks to revised main text.	Main Text Line 475
3.5	Figure 3: please explain the blue shaded areas in the caption.	Added clarifying text to Fig. 3 caption	Main Text Line 608, 615
3.6	Figure S6: It is not quite clear to me what “aerobic to anaerobic” and “anaerobic to aerobic transitional growth curves” means? Just aerobic and anaerobic growth curves? Please explain better.	Added clarifying text to revised supplementary information	SI Text Line 102

Reviewers' Comments:

Reviewer #1:

Remarks to the Author:

I have no further concerns. Great job to the authors!

Sarah Glaven

Reviewer #2:

Remarks to the Author:

The authors have addressed all comments adequately and I recommend this manuscript should be accepted.